



# Data reduction for inverse modeling: an adaptive approach v1.0

Xiaoling Liu[1], August L. Weinbren[1], He Chang[1], Jovan Tadić[2], Marikate E. Mountain[3], Michael E. Trudeau[4], Arlyn E. Andrews[4], Zichong Chen[1], and Scot M. Miller[1]

[1]Department of Environmental Health and Engineering, Johns Hopkins University, Baltimore, MD, USA
[2]Lawrence Berkeley National Laboratory, Berkeley, CA, USA
[3]Atmospheric and Environmental Research, Inc., Lexington, MA, USA
[4]Global Monitoring Laboratory, National Oceanic and Atmospheric Administration, Boulder, CO, USA

**Correspondence:** Scot M. Miller (smill191@jhu.edu)

**Abstract.** The number of greenhouse gas (GHG) observing satellites has greatly expanded in recent years, and these new datasets provide an unprecedented constraint on global GHG sources and sinks. However, a continuing challenge for inverse models that are used to estimate these sources and sinks is the sheer number of satellite observations, sometimes in the millions per day. These massive datasets often make it prohibitive to implement inverse modeling calculations and/or assimilate the
observations using many types of atmospheric models. Although these satellite datasets are very large, the information content of any single observation is often modest and non-exclusive due to redundancy with neighboring observations and due to measurement noise. In this study, we develop an adaptive approach to reduce the size of satellite datasets using geostatistics. A guiding principle is to reduce the data more in regions with little variability in the observations and less in regions with high variability. We subsequently tune and evaluate the approach using synthetic and real data case studies for North America from
NASA's Orbiting Carbon Observatory-2 (OCO-2) satellite. The proposed approach to data reduction yields more accurate $CO_2$ flux estimates than the commonly-used method of binning and averaging the satellite data. We further develop a metric for choosing a level of data reduction; we can reduce the satellite dataset to an average of one observation per $\sim 80 - 140$ km for the specific case studies here without substantially compromising the flux estimate, but we find that reducing the data further quickly degrades the accuracy of the estimated fluxes. Overall, the approach developed here could be applied to a range of
inverse problems that use very large trace gas datasets.

## 1 Introduction

Satellite observations of greenhouse gases (GHGs) have dramatically expanded over the past decade. New satellites with smaller footprints, wider viewing angles, and efficient scanning can collect millions of observations per day at high density and with broad spatial coverage. Remote sensing of carbon dioxide ($CO_2$) is a prime example. The Greenhouse Gases Observing
Satellite (GOSAT), launched in early 2009, is the first satellite dedicated to observing $CO_2$ and methane ($CH_4$) from space. GOSAT collects a modest $\sim 1 \times 10^3$ cloud-free soundings or observations per day. The Orbiting Carbon Observatory 2 (OCO-2) launched five years later in late 2014 and is NASA's first satellite dedicated to observing $CO_2$. It collects far more cloud-free soundings than GOSAT – on the order of $1 \times 10^5$ (Crisp, 2015; Eldering et al., 2017). By contrast, NASA's forthcoming





Geostationary Carbon Observatory (GeoCarb), planned for launch in the early 2020s, is slated to collect $\sim 1 \times 10^7$ soundings
each day (Buis, 2018). A substantial fraction of these soundings will be unusable due to cloud contamination, but GeoCarb can
reduce contamination by scanning cloud-free regions.

These satellites observe average $CO_2$ mixing ratios across a vertical column of the atmosphere ($XCO_2$), and these $XCO_2$
measurements can be used to estimate surface $CO_2$ fluxes using inverse modeling. Specifically, an inverse model will combine
satellite observations ($z$, dimensions $n \times 1$) with an estimate of atmospheric transport ($\mathbf{H}$, $n \times m$) to estimate surface fluxes ($s$,
$m \times 1$):

$$z = \mathbf{H}s + \epsilon, \tag{1}$$

where $\epsilon$ ($n \times 1$) is a vector of errors in the measurements and atmospheric modeling system. The objective of the inverse
model is to estimate $s$ given $z$ and $\mathbf{H}$. Most existing inverse models also require an estimate of the statistical properties of
$\epsilon$ to ensure that the solution does not under- or over-fit the atmospheric observations ($z$). There are many different strategies
for estimating the fluxes ($s$); most existing studies implement an inverse model that is based upon Bayesian statistics. Refer
to Rodgers (2000), Michalak et al. (2004), and Brasseur and Jacob (2017) for an overview of commonly-used strategies for
inverse modeling.

Large satellite datasets often pose computational problems for inverse modeling, specifically for calculations that involve
the atmospheric model, $\mathbf{H}$. The associated challenges vary depending upon the the type of atmospheric model. For example,
one approach to inverse modeling is to estimate $\mathbf{H}$ using a particle-following model, also known as a back-trajectory model.
These models estimate how fluxes or emissions from different regions would impact mixing ratios at a downwind observation
site. Commonly-used models include the Stochastic Time-Inverted Lagrangian Transport (STILT) model (Lin et al., 2003;
Nehrkorn et al., 2010), the FLEXible PARTicle dispersion model (FLEXPART) (Pisso et al., 2019), and the Hybrid Single-
Particle Lagrangian Integrated Trajectory model (HYSPLIT) (Stein et al., 2016). One must run simulations of the particle-
following model for each of $n$ observations used in the inverse model, and each simulation becomes a different row of $\mathbf{H}$.
As a result of this setup, the computational cost of the particle-following model scales with the number of observations, the
size of the modeling domain, and the resolution of the model. This approach is commonly employed for ground- and aircraft-
based atmospheric observations but can quickly become computationally challenging for large satellite datasets (e.g., Wu et al.,
2018).

Another common approach to atmospheric modeling is to use the forward and adjoint of a gridded atmospheric model, also
known as an Eulerian model. Common models include the Goddard Earth Observing System – Chemistry model (GEOS-
Chem) (e.g., Henze et al., 2007; Liu et al., 2017) and TM5 (e.g., Krol et al., 2005; Bergamaschi et al., 2005). These models
are not typically used to explicitly calculate $\mathbf{H}$ (the forward model) or $\mathbf{H}^T$ (the adjoint). Rather, these models are often used
to calculate the product of $\mathbf{H}$ or $\mathbf{H}^T$ and a vector. One can then estimate $s$ by iterating toward the minimum of an objective
function using a series of matrix-vector products that involve $\mathbf{H}$ and $\mathbf{H}^T$ (e.g., Brasseur and Jacob, 2017). Large satellite
datasets can also create computational challenges for inverse models that follow this approach. The cost of running the forward
and adjoint models usually increases as the number of observations increase, particularly for the model adjoint. For example,





a single run of the GEOS-Chem adjoint model requires approximately 30 days of wall clock time for a global resolution of 2° latitude by 2.5° longitude and a year of OCO-2 observations from the "lite" file. Most inverse models require many, 60  sequential runs of the adjoint model, yielding a total computing time that would be absolutely prohibitive. In addition, many adjoint models for GHG applications are not designed to exploit parallel computing architecture. However, parallel versions are currently under development at the time of writing (e.g., Eastham et al., 2018).

The most common solution to date for these computational problems has been to reduce the size of the satellite dataset. One approach has been to bin and average the data across a set interval and/or run the atmospheric model at a set interval 65  along the satellite flight track. For example, recent inverse models for OCO-2 use data that has been binned and averaged every 10-seconds along the satellite flight track (e.g., Crowell et al., 2019). This approach yields approximately one observation per 70 km, far fewer observations than the original OCO-2 dataset.

Relatedly, scientists that use a trajectory model for atmospheric simulations will often run the model at a set interval along the flight track. For example, scientists at NOAA have generated trajectory simulations for OCO-2 data over North America 70  as part of the CarbonTracker-Lagrange project (e.g., NOAA Global Monitoring Laboratory, 2020a; Miller et al., 2020). These runs have been generated for a single location every 2-seconds along each satellite flight track, thereby reducing the number of model simulations required.

These existing strategies for data reduction present several challenges. First, one must decide how frequently to average the data (i.e., across how many seconds or kilometers) or how frequently to generate atmospheric simulations along a satellite 75  flight track. It is not alway practical to re-run the atmospheric model and inverse model with different levels of data reduction to decide on an optimal approach – due to the computing time involved. Instead, this decision is often based upon the spatial resolution of the atmospheric transport model (e.g., NOAA Global Monitoring Laboratory, 2020a) or the anticipated spatial resolution of the flux estimate (e.g., Crowell et al., 2019). That approach may become challenging for new satellites with frequent repeat sampling, like GeoCarb, and as new, high resolution meteorological products or atmospheric simulations become 80  available. For example, in test simulations, the GEOS-Chem $CH_4$ and CO adjoint models for the nested North American domain (0.25° × 0.33° resolution) required between 100 and 120 minutes to run for a single day of TROPOMI observations, even after averaging the observations to the resolution of the model grid. Second, one level of averaging or data reduction (e.g., 2-second versus 10-second) may work better for one inverse modeling setup or one satellite dataset than another, depending upon the spatial and temporal resolution of the inverse model and the spatial and temporal properties of the GHG of inter-85  est, among other factors. Lastly, satellite observations are typically non-stationary: they exhibit different spatial and temporal variability in different locations and in different seasons (e.g., Katzfuss and Cressie, 2011; Hammerling et al., 2012a). These differences may be important to account for when reducing the size of the satellite dataset. For example, OCO-2 observations collected over the remote ocean have a lower variance and are correlated across longer distances than observations collected over terrestrial regions with heterogeneous surface sources and sinks (e.g., Eldering et al., 2017). A one-size-fits-all approach 90  to data reduction may not be ideal in this circumstance. Instead, it may be advantageous to reduce the size of the dataset more in regions with little variability and less in regions with greater variability.



Scientists in other academic disciplines have also grappled with many of these challenges, albeit in the context of very different scientific applications. For example, data reduction has become common in computer graphics and data visualization because many remote sensing and/or medical images are too large to render and display at native resolution (e.g., Li et al., 2018). Numerous studies reduce the size of the image through a process known as mesh reduction; these algorithms reduce the mesh more in regions of the image with little variability and less in locations with high variability (e.g., Schroeder et al., 1992; Garland and Heckbert, 1997; Brodsky and Watson, 2000; Li et al., 2018). The algorithms are therefore also adaptable to different images.

Data reduction has also become common in weather data assimilation, where the reduced datasets are typically referred to as "superobservations" or "superobs". In most existing meteorology studies, the data is divided into different grid boxes and averaged, analogous to the approach used in recent GHG studies (e.g., Lorenc, 1981; Miyoshi and Kunii, 2012). More recently, however, several studies have proposed "adaptive" or "intelligent" approaches to data reduction (e.g., Ochotta et al., 2005; Ramachandran et al., 2005; Lazarus et al., 2010; Richman et al., 2015). These studies preferentially reduce or thin the data more in regions where the observations have little variability or provide redundant information. Existing studies have used different algorithms to attain this goal, including mesh reduction (Ramachandran et al., 2005), data clustering (e.g., Ochotta et al., 2005), and machine learning (Richman et al., 2015). Compared to these meteorology studies, data reduction in atmospheric inverse modeling presents unique challenges. In weather data assimilation, observations are used to directly nudge or adjust a weather model in adjacent grid boxes. In inverse modeling, by contrast, the atmospheric observations and unknown GHG fluxes are fundamentally different quantities with complex relationships determined by atmospheric winds.

In the present study, we develop an approach to data reduction for inverse modeling of GHG observations. This approach follows the principles of adaptive reduction: we reduce the $XCO_2$ data more in regions with little variability in the observations and less in regions with high variability. The goals of this approach are two-fold. First, improve the computational feasibility of inverse modeling using satellite data while preserving the accuracy of the estimated fluxes. Second, develop an objective means to decide on the optimal level of data reduction for a given satellite dataset and a given inverse modeling problem. We subsequently tune and evaluate this approach using several case studies from the OCO-2 satellite – case studies that use synthetic and real data and case studies from different seasons of the year. We then compare $CO_2$ fluxes estimated using the proposed approach against fluxes estimated using a satellite dataset that has been averaged to reduce its size. This comparison provides a lens to evaluate the costs and benefits of the proposed approach to data reduction versus the commonly-used approach of averaging the data. The approach described here is designed not only for OCO-2 but could be applied to current and future observations of $CO_2$ (e.g., from GeoCarb) and observations of $CH_4$ (e.g., from the TROPOspheric Monitoring Instrument (TROPOMI) and GeoCarb).

## 2 Approach to data reduction

We develop an approach to data reduction for inverse modeling that leverages tools from geostatistics. Geostatistical tools, like variogram modeling and kriging, have become widespread in spatial data analysis (e.g., Kitanidis, 1997; Wackernagel, 2003),





and these tools are often straightforward to implement using software packages in R, Matlab, Python, and other scientific programming languages. Furthermore, geostatistical tools are already used throughout inverse modeling and therefore offer an appealing framework for data reduction.

The overall strategy developed here is to first characterize the spatial properties of the observations using variogram analysis and second, use kriging to interpolate the satellite observations to a number of locations that is smaller than the original dataset.

The choice of locations is informed by the variogram analysis: we retain fewer locations in regions where the observations are correlated over longer distances and more locations in regions with a shorter decorrelation length.

### 2.1 Step 1: Evaluate the spatial properties of the satellite data

We estimate the degree of spatial correlation in the satellite observations using a variogram analysis (e.g., Kitanidis, 1997). This analysis yields an estimate of the decorrelation length – the distance at which the correlation between any two observations is

effectively zero.

In this study, we estimate the decorrelation length by creating a variogram of the satellite observations (Fig. S6). A variogram is a geostatistical tool that is used to quantify the differences among observations as a function of distance. The variogram of the observations is known as an empirical variogram, and we then fit a model to this empirical variogram using a least squares fit to estimate the decorrelation length (e.g., Kitanidis, 1997; Wackernagel, 2003). There are many possible choices for a variogram

model, and we choose an exponential model because it has been used in several existing studies of satellite-based $XCO_2$ observations (e.g., Hammerling et al., 2012a; Zeng et al., 2014; Guo et al., 2015; Tadić et al., 2015, 2017). The covariances between observations in this model decay exponentially as a function of distance. Note that the exponential model yields an estimate of the e-folding distance, the distance at which covariances decay by a factor of $e$. In this study, we report the decorrelation length ($l$) or three times the e-folding distance; this is the distance at which the covariances effectively decay to

zero. Refer to Kitanidis (1997) or Wackernagel (2003) for a review of different variogram models and model fitting.

We specifically estimate the decorrelation length along individual satellite flight tracks and estimate different lengths at different locations along each track. The spatial properties of the satellite observations often differ in different regions of the globe, and these differences are important to account for. In this particular study, we estimate a new decorrelation length every two seconds along the flight track and include all observations within 2000 km when making the estimate at each location

(as in Hammerling et al., 2012a, b). Note that we do not quantify correlations or covariances among different flight tracks or different days for the case studies using OCO-2 (Sect. 3); that satellite has a narrow swath of $\sim 10$ km and a 16 day revisit time, so the individual flight tracks on a given day or week are spaced relatively far apart (e.g., Crisp, 2015; Eldering et al., 2017). For new and forthcoming satellites with a wider swath and/or more frequent revisit time, one could quantify zonal, meridional, and/or temporal decorrelation lengths, depending upon the characteristics of the satellite in question.

### 155 2.2 Step 2: Reduce the data using kriging

We subsequently reduce the satellite dataset by estimating atmospheric $CO_2$ at one location per fraction of a correlation length along each satellite flight track. For example, a modeler could reduce the dataset to one observation per $0.1l$ or $1.0l$. The latter





choice would reduce the size of the dataset to a much greater degree but increase the risk of losing information that would ultimately inform the inverse model. Sect. 2.3 discusses strategies for deciding on an optimal level of data reduction.

The correlation length will differ in different locations, and this procedure will therefore yield a different density of points in different regions. For example, the proposed approach will result in a greater density of points in regions where $XCO_2$ varies across small spatial scales and a lower density of points in regions where $XCO_2$ is correlated across long distances.

This approach is conceptually similar to several adaptive strategies for data reduction in other scientific fields. Many existing studies either remove, merge, or cluster data points based on spatial variability. In computer visualization, mesh reduction

studies merge or remove vertices from the image based upon the curvature or flatness of the original image, and different studies use various metrics to quantify this curvature and flatness (e.g., Schroeder et al., 1992; Garland and Heckbert, 1997; Brodsky and Watson, 2000; Li et al., 2018). Studies in meteorology use similar algorithms. For example, Ochotta et al. (2005) developed a metric to cluster observations based upon both the squared distance between observations and the squared difference in the observation values. Similarly, Ramachandran et al. (2005) reduced the data based upon the variance of the data in each locale.

In this particular study, we use the decorrelation length, a common tool in geostatistics, to quantify the variability of the original data and guide the data reduction.

At each chosen location along the flight track, we subsequently interpolate the observations using ordinary kriging (e.g., Kitanidis, 1997). Numerous existing studies have applied various forms of kriging to interpolate satellite observations of $CO_2$ (e.g., Katzfuss and Cressie, 2011; Hammerling et al., 2012a, b; Tadić et al., 2015). Kriging accounts for the spatial and/or

temporal properties of the quantity of interest, yielding a more accurate estimate. Kriging also yields an estimate of uncertainty in the estimated quantity, in this case uncertainties in the reduced $XCO_2$ dataset. These uncertainties account for the variability (or lack therefore) of the OCO-2 data in the vicinity of each location, the density or sparsity of the original OCO-2 dataset near each location, and random noise in the OCO-2 observations, among other error sources. The uncertainties estimated from kriging could be used to inform the covariance matrices in an inverse model or to help evaluate the information in the reduced

dataset. Both the best estimate of $XCO_2$ and the corresponding uncertainties can be calculated using a simple linear system of equations.

We specifically implement ordinary kriging using a moving neighborhood (e.g., Kitanidis, 1997; Hammerling et al., 2012a, b); the quantity of interest is only estimated at a single location or a subset of locations at one time using nearby observations. This approach contrasts with other variants of kriging that incorporate all observations to estimate all unknown locations

simultaneously. Ordinary kriging with a moving neighborhood is particularly useful when the observations are non-stationary and exhibit different spatial properties and/or error characteristics in different regions, as is often the case with $XCO_2$ (e.g., Hammerling et al., 2012a).

Ordinary kriging with a moving neighborhood requires two steps. First, a modeler must estimate the spatial properties of the observations in the vicinity of the estimation location. We estimate these properties as part of the analysis in Sect. 2.1 and

use that estimate as an input in ordinary kriging. Second, we estimate $XCO_2$ at each location of interest by solving a system of linear equations. Kitanidis (1997) describes this approach in detail, and Hammerling et al. (2012a), Hammerling et al. (2012b)





describe the application of moving neighborhood kriging to observations of $XCO_2$. The $XCO_2$ estimates from kriging can then be incorporated as observations in inverse modeling.

Note that traditional kriging models are designed to interpolate the quantity of interest to the same spatial support as the observations. In other words, the footprint size of the kriging estimate will be the same as that of the observations. This setup works well for the case study presented here; the atmospheric simulations in this study are generated using a particle-following model, and each simulation corresponds to a specific point location and time along an OCO-2 flight track. By contrast, a variant of kriging known as block kriging can be used to estimate a representative or average value for an entire grid box (e.g., Wackernagel, 2003; Tadić et al., 2015, 2017). This approach may be desirable when generating atmospheric simulations using an Eulerian model where the outputs represent grid averages. Tadić et al. (2015) and Tadić et al. (2017) describe this approach in detail, including applications to interpolating $XCO_2$.

## 2.3 Step 3: Decide on an optimal level of data reduction

A modeler must decide on an optimal level of data reduction. That decision is often based on multiple considerations – the native spatial resolution of the atmospheric model, the computational demands of the inverse model, and the accuracy of the resulting flux estimate. For example, the resolution of the atmospheric model may help dictate a level of data reduction. An atmospheric model will not be able to resolve patterns in the fluxes at spatial scales smaller than the model resolution, so there may be little need to assimilate $CO_2$ observations at finer density than the model resolution.

For the specific algorithm described here, one must decide on a fraction of a correlation length and reduce the dataset accordingly. A modeler could decide on the optimal level of data reduction using a brute force approach: create numerous datasets with different levels of data reduction, run the inverse model on each, and decide on a level of data reduction based upon a comparison of the estimated fluxes. In practice, this approach is time consuming.

Instead, we propose a criterion for choosing a level of data reduction based upon the variance of the satellite data. Specifically, we calculate the variance of the original OCO-2 observations between each of the reduced data points. After calculating the variance between each pair of reduced data points, we then average all of the calculated variances – across all pairs or intervals. This variance represents the variability in the data that is lost through the process of data reduction, and it provides a metric for choosing a level of data reduction that does not require re-running the inverse model. This number is smallest when the data reduction is minimal and increases for greater levels of data reduction. For the case studies in Sect. 3, this variance is often a non-linear function of the level of data reduction; it increases slowly if the data reduction is minimal, reaches an inflection point, and then increases more quickly at greater levels of data reduction. A modeler can then choose a level of data reduction that is preferably below the inflection point and therefore reduces the potential for information loss while balancing the computational requirements of the inverse model.

We evaluate this proposed approach for deciding on a level of data reduction through several case studies based upon the OCO-2 satellite, described in detail in the next section (Sect. 3).





## 3 Description of the case studies

We evaluate the data reduction algorithm using three case studies based on the OCO-2 satellite. In each case, we estimate $CO_2$ fluxes across North America for six weeks at a 3-hourly temporal resolution and a $1° \times 1°$ latitude-longitude spatial resolution. Note that this setup targets a particular application of OCO-2 observations to inverse modeling across a continent. One could apply data reduction to inverse models that target urban areas or the entire globe, but the algorithm tuning (e.g., Sect. 2.3) and inverse modeling results will depend upon the particular application involved.

We specifically estimate fluxes using synthetic observations from July and early August 2015, using synthetic observations from March and early April 2015, and using real observations from July and early August 2015. Synthetic observations make it possible to compare the results against a known solution; they are therefore particularly useful for the evaluating the data reduction algorithm proposed here. We further evaluate the algorithm in a real data simulation that mirrors real-world inverse modeling applications. Note that we present the details of the summer real data case study in the SI and focus on the synthetic

case studies in the main text. The results of the real data case study are qualitatively very similar to the synthetic case studies, so we include that information in the SI to avoid duplicating similar information in the main text.

We further estimate the $CO_2$ fluxes using geostatistical inverse model (e.g., Kitanidis and Vomvoris, 1983; Michalak et al., 2004; Miller et al., 2020). The inverse model used here also has a non-informative prior. In other words, the prior has no spatial variability (e.g., Michalak et al., 2004; Mueller et al., 2008). As a result, any patterns in the estimated fluxes reflect the

information content of the observations, not any prior information. This setup is identical to the case studies in Miller et al. (2020), and the reader is referred to both the SI and that study for additional detail.

The case studies here also use atmospheric transport simulations from NOAA's CarbonTracker-Lagrange project (e.g., Hu et al., 2019; NOAA Global Monitoring Laboratory, 2020a). These simulations are generated using the Weather Research and Forecasting (WRF) model coupled with the Stochastic Time-Inverted Lagrangian Transport Model (STILT) model (e.g., Lin

et al., 2003; Nehrkorn et al., 2010). The simulations have a spatial resolution of 10 km over most of the Continental US and a resolution of 30 km across other regions of North America. Miller et al. (2020) provides additional detail on the specific setup of the WRF-STILT runs used here. Note that the STILT simulations for CarbonTracker-Lagrange were generated every two seconds along the OCO-2 flight track and not at every individual OCO-2 observation due to the large number of observations and due to computational constraints. Hence, we only evaluate data reduction that yields fewer than one observation every two

seconds for the case studies here.

We further create the synthetic data for each case study using WRF-STILT and $CO_2$ fluxes from NOAA's CarbonTracker (CT2017) product (Peters et al., 2007; NOAA Global Monitoring Laboratory, 2020b). The synthetic $CO_2$ fluxes not only include biospheric fluxes but also anthropogenic and biomass burning emissions. The synthetic observations also include noise ($\epsilon$) that is added to simulate measurement and atmospheric modeling errors. For the summer case studies here, these errors

have a variance of $(2 \text{ ppm})^2$ (as in Miller et al., 2020). We also include error covariances to account for spatial correlation among these errors. We use decorrelation length from Kulawik et al. (2019), who quantified errors in OCO-2 observations and estimated a decorrelation parameter of $0.3°$ using an exponential variogram model. In the winter case study, we use a





slightly smaller error variance of $(1.5 \text{ ppm})^2$ because there is less regional variability in atmospheric $CO_2$ in winter. Note that we only include land nadir and land glint observations in the case studies and exclude ocean glint observations because those
observations have known biases (O'Dell et al., 2018).

## 4    Results and discussion

### 4.1    Spatial properties of the OCO-2 observations

We estimate correlation lengths that are generally longer in winter when biospheric fluxes are small than in summer when there are large spatial and temporal variations in biospheric fluxes. Figure 1 displays the estimated correlation lengths along
the OCO-2 flight tracks for the summer (a) and winter (b) case studies. Most of the estimated correlation lengths range from ~250 km to 1000 km. Note that there are likely multiple different scales of variability in the OCO-2 observations: fine-scale variability due to retrieval errors (e.g., Kulawik et al., 2019; O'Dell et al., 2018), small-scale variability due to variations in mesoscale meteorology (e.g., Torres et al., 2019), and broad variability due to synoptic meteorology and regional patterns in $CO_2$ fluxes. We specifically focus on quantifying synoptic scale variability in Fig. 1 – because the objective of the case studies
is to estimate broad, regional patterns in $CO_2$ fluxes across an entire continent.

The analysis in Fig. 1 also indicates substantial heterogeneity in the correlation lengths. Correlation lengths are often similar along a single flight track but vary among different tracks. These differences between flight tracks are most likely due to a combination of variations in synoptic meteorology and variability in the underlying $CO_2$ fluxes. Indeed, several studies have shown that meteorological variability can explain a substantial fraction of variability in $XCO_2$ across different spatial scales
(e.g., Parazoo et al., 2008; Keppel-Aleks et al., 2011; Torres et al., 2019).

Two flight tracks that cross California, Oregon, and Washington illustrate the likely impacts of fluxes and meteorology on heterogeneity in the synthetic OCO-2 observations. One track, on July 16, 2015, exhibits relatively little variability in $XCO_2$, and we estimate an average correlation length of 794 km along the track with a standard deviation of 266 km. By contrast, another nearby track from July 21 exhibits far more $XCO_2$ variability, and we estimate a shorter mean correlation length of
221 km with a standard deviation of 53 km along the track. These large differences likely reflect differences in the underlying $CO_2$ fluxes and in meteorology on the respective days. The July 16 track passes through eastern California, Nevada, Eastern Oregon, and Eastern Washington, desert regions with little heterogeneity in $CO_2$ fluxes. By contrast, the track from July 21 passes over the Sierra Nevada mountains and over multiple heterogeneous biome types (e.g., desert and temperate rainforest). Furthermore, weather maps indicate that a cold front passed through the Pacific Northwest on July 21 with variable winds
on either side of the front (NOAA National Centers for Environmental Prediction Weather Prediction Center, 2020). These differences in transport and surface fluxes likely yield very different patterns of variability in two satellite flight tracks that are geographically close to one another.



## 4.2 Estimated CO$_2$ fluxes using the reduced OCO-2 dataset

The data reduction approach proposed here yields flux estimates that faithfully reproduce patterns in the synthetic CO$_2$ obser-

vations. With that said, data reduction is always a compromise between the accuracy of the flux estimate and the computational requirements of the inverse model. As such, the accuracy of the flux estimate begins to degrade at high levels of data reduction. Figures 3 and 4 summarizes many of these features and displays maps of the time-averaged fluxes from the summer and winter case studies, respectively. The first panel (a) in each figure contains the CarbonTracker fluxes that were used to generate the synthetic OCO-2 observations. The second panel (b) shows the fluxes estimated using the full, synthetic OCO-2 dataset with

no reduction; the estimated CO$_2$ fluxes shown in these panels do not have the same level of spatial definition as the original CarbonTracker fluxes (panel a), but the estimates broadly reproduce the spatial patterns in CarbonTracker. The inverse model in this study uses a non-informative prior, so any patterns in panel (b) are solely informed by the observations and not the result of prior flux information. The patterns in panel (b) indicate that the synthetic OCO-2 observations can be used to recover continental-scale spatial features in the fluxes, but the observations and inverse model do not have the sensitivity or information

content to recover more detailed features. Subsequent panels (c) display the fluxes estimated using observations that have been reduced to a modest level – one observation per $0.2l$ or an average of one observation per 100 km for the summer case study and 140 km for the winter case study. The final panel in each figure displays a severe level of reduction – one observation per $0.75l$, an average of one observation per 400 km for the summer case study and 540 km for the summer case study.

In each case, the fluxes using the $0.2l$ data (c) reproduce spatial patterns in the fluxes estimated with no reduction (b).

By contrast, the fluxes estimated using the $0.75l$ data lack spatial definition, and are therefore not an ideal estimate of the synthetic fluxes (a). Note that we also conducted a real data case study for summer 2015. Those results have broadly similar characteristics to the synthetic data case study and are discussed in detail in the SI.

In both of the case studies, the data reduction approach proposed here yields more accurate flux estimates than binning and averaging the observations. We reduce the data using both averaging and the geostatistical approach proposed in this study.

We subsequently estimate fluxes using the reduced datasets and compare the results against the fluxes estimated using the full dataset without any data reduction. Figure 5 displays the results of this analysis – the root mean squared error (RMSE) of the grid-scale, 3-hourly estimated fluxes relative to fluxes calculated from the full dataset. In both the winter and summer case studies and at almost all levels of data reduction, the geostatistical approach produces fluxes with a lower RMSE.

Note that all results in Fig. 5 display a clear inflection point: the RMSE is relatively low at low levels of data reduction and

increases rapidly at high levels of reduction. The chosen level of data reduction should be at or below this inflection point, or the inverse model will yield an inaccurate flux estimate. The inflection point for the geostatistical approach occurs at a higher level of data reduction than for data averaging. In other words, the RMSE for the geostatistical approach remains low at a greater degree of data reduction than for averaging.

The RMSE, however, may not be the only criteria to consider when deciding on a level of data reduction. Specifically, the

spatial patterns in the fluxes begin to degrade at a lower level of data reduction than the RMSE. For example, in both the winter, summer, and real data case studies, the monthly-averaged fluxes begin to lose spatial definition at data reduction levels greater





than $0.15l$ to $0.2l$ (an average of one observation per 80-100 km for the summer case study and one observation per 100-140 km for the winter case study). Hence, it may be advisable to balance multiple criteria when deciding on an optimal level of data reduction, depending upon the goals of the inverse modeling study.

### 4.3   Determining an optimal level of data reduction

In the previous section (Sect. 4.2), we evaluate the data reduction by comparing the resulting estimates of $CO_2$ fluxes. However, this approach may not work well if the inverse model is time-consuming and/or computationally intensive, as is often the case for satellite-based inverse modeling. For example, in Fig. 5 we run the inverse model twenty times for each case study to estimate the fluxes using different levels of data reduction and evaluate the results against fluxes estimated using the original, synthetic OCO-2 data. An inverse model can take days to run using large satellite datasets, so it may not be feasible or desirable to run the inverse model numerous times. Furthermore, the original satellite dataset (i.e., no reduction) is often too large to run through an inverse model, making it impossible to conduct the type of comparison in Figs. 3 and 4. For example, an inverse model using a full year of global OCO-2 data and the GEOS-Chem adjoint model could take months to run (Sect. 1).

Instead, we propose an approach to evaluate the data reduction in a way that does not require re-running the inverse model (Sect. 2.3). This approach is based upon the variance of the original data between each of the reduced data points; this variance is a measure of the variability in the data that is lost through the process of data reduction.

In each of the OCO-2 case studies, this metric provides a reasonable and informative means to decide on an appropriate level of data reduction. Figure 6 displays this metric calculated for the summer (a) and winter (b) case studies at multiple different levels of data reduction. The variance lost through the process of data reduction is lowest at small levels of data reduction and increases non-linearly at higher levels of data reduction. The summer case study (a) is highly non-linear and reaches a very clear inflection point. By contrast, the winter case study (b) does not have as clear of an inflection point, but the variance does increase more quickly at higher levels of data reduction.

This metric also mirrors many of the patterns in the flux maps (Figs. 3 and 4) and RMSE calculations (Fig. 5). For example, fluxes in the summer case study loose spatial definition at data reduction levels greater than $0.2l$ (equivalent to 2263 data points). Indeed, the variances in Fig. 6a begin to increase more rapidly after at data reduction levels greater than $0.2l$. By contrast, the fluxes in the winter case study progressively lose spatial definition, but that loss is particularly notable at high levels of reduction. That pattern is similar to the pattern in the variances in Fig. 6b. Furthermore, the patterns in Fig. 6 also mirror many of the patterns in the RMSE (Fig. 5). Both the RMSE and variances for the summer case study reach an inflection point between 2000 and 1000 observations, at which point both begin to increase rapidly. The RMSE and variance plots for the winter case study do not look identical (Figs. 5b and 6b). With that said, the pattern in the winter case study looks qualitatively more akin to the degradation of spatial patterns in the plotted fluxes than it does to the RMSE in Fig. 5b.

At the end of the day, it is arguably difficult to identify a single metric for deciding on an appropriate level of data reduction, and the right metric may depend upon the goal of the inverse model (e.g., identifying spatial patterns, temporal patterns, and/or flux totals). With that said, the metric proposed in this section is a computationally efficient option that summarizes many of the features of data reduction described in the previous sections (e.g., Sect. 4.2).





## 4.4   Computational costs

We find that the computational costs of the data reduction algorithm are reasonable for the case studies explored here and are far less than the computing time associated with generating atmospheric model simulations. Both the variogram analysis (Sect. 2.1) and kriging (Sect. 2.2) are implemented using a moving neighborhood, thereby limiting the number of observations included

in any given variogram or kriging calculation and reducing computing time. For example, we use a moving neighborhood with a radius of 2000 km for the case studies (as in Hammerling et al. (2012a)), approximately half the width of the continental United States. Each empirical variogram required an average of 0.05 s to calculate using the R programming language, and each kriging estimate required an average of 0.02 s. By contrast, a single STILT model simulation corresponding to a single OCO-2 observation required far more computing time – at least 20 minutes (Sect. 3).

Furthermore, one can distribute the variogram and kriging calculations across multiple computing cores and nodes, reducing the required computing time. Specifically, the calculation of each decorrelation length and each kriging estimate is independent of every other calculation or location, so these individual calculations can be spread across as many cores or nodes as desired. There is also some flexibility in the implementation of this algorithm and therefore in its computational cost. For example, we calculate the variogram every two seconds along the OCO-2 flight track, but one could calculate the variogram at less

frequent intervals. For example, Hammerling et al. (2012a) implemented moving neighborhood kriging for synthetic OCO-2 observations and calculated variogram parameters for each location on a $1°$ latitude by $1.25°$ grid. One can also define the moving neighborhood differently with computational considerations in mind. For example, Tadić et al. (2015) and Tadić et al. (2017) limit the moving neighborhood to 500 $XCO_2$ observations. Instead of including all observations within a given radius, they choose observations to include in the moving neighborhood using a randomized algorithm, and the algorithm preferentially

chooses observations that are closer to the estimation location over observations that are far away. That strategy yields accurate variogram parameters and kriging estimates while ensuring that the number of observations within a moving neighborhood is not so large as to pose a computational burden.

## 5   Conclusions

In many instances, new satellite datasets are simply too large to assimilate in an inverse model given the current computational

limitations of existing atmospheric models. In an ideal world, it would be possible to assimilate all available GHG observations to exploit the full information content of these massive new satellite datasets. However, that ideal is not computationally feasible in many instances, and modelers often need a strategy to reduce the size of these datasets. At minimum, this strategy should reduce the computational demands of inverse modeling while yielding flux estimates that accurately reproduce key information on the magnitude and distribution of surface fluxes. A complicating factor is that satellite observations often

exhibit very different variability in different regions and/or on different days – depending on factors like regional variability in GHG fluxes and variations in meteorology. In this work, we argue that a data reduction strategy should account for this variability and that doing so typically yields a more accurate flux estimate.

One could develop a strategy for data reduction using many different statistical and mathematical tools, and we specifically develop a strategy using geostatistics because it provides a convenient way to quantify and account for the spatial variability

of the satellite observations. In the case studies presented here based on NASA's OCO-2 satellite, this strategy outperforms data averaging, a common and straightforward approach to data reduction but one that does not account for the variable spatial properties of the observations. The specific implementation of this strategy will likely vary depending upon the satellite dataset in question and the specifics of the atmospheric model. To that end, we also develop and evaluate a computationally efficient metric to help choose an appropriate level of data reduction – a metric that does not require re-running the inverse model

numerous times.

Future improvements to atmospheric models (e.g., the ability to better exploit parallel computing) and increased access to high performance computing resources will hopefully make it possible to implement inverse modeling with larger and larger atmospheric datasets while minimizing the need for data reduction. With that said, forthcoming satellites like NASA's GeoCarb mission promise to collect unprecedented numbers of atmospheric GHG observations, and these new missions may make data

reduction more necessary than ever.

*Code and data availability.*   The code used in this study for data reduction is available on Github at https://doi.org/10.5281/zenodo.3899307. The inverse modeling code used in the case studies is also available at https://doi.org/10.5281/zenodo.3241524, and the model simulations used to construct the summer case study are available on Zenodo at https://doi.org/10.5281/zenodo.3241466.

*Author contributions.*   X.L. A.L.W. J.T. and S.M.M. designed the study. X.L. A.L.W. H.C., Z.C. and S.M.M. conducted the analysis. M.M.,

M.T. and A.A. developed the strategy for simulating OCO-2 soundings using WRF-STILT and provided the footprints. S.M.M. drafted the manuscript, and all authors helped edit the manuscript.

*Competing interests.*   The authors declare they have no competing interests.

*Acknowledgements.*   We thank Thomas Nehrkorn (Atmospheric and Environmental Research (AER), Inc.) for help with the atmospheric modeling simulations used in this study. We also thank Amy Braverman (NASA) for her advice and input on the project.

*Financial support.*   This work is supported by NASA ROSES grant no. 80NSSC18K0976. CarbonTracker-Lagrange footprint production was supported by NASA the NASA Carbon Monitoring System via interagency agreement NNH14AY37I.



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





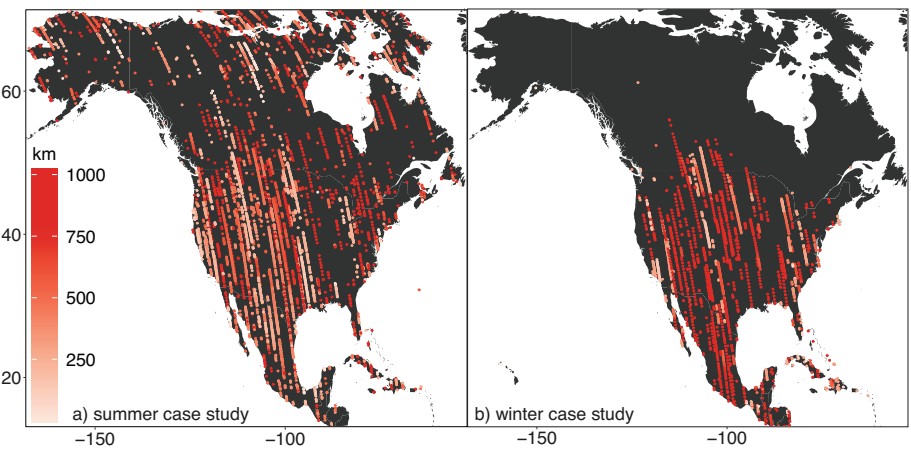

**Figure 1.** Correlation lengths estimated along OCO-2 flight tracks for (a) the summer synthetic case study and (b) the winter synthetic case study. The estimated correlation lengths are typically shorter in summer when biospheric fluxes exhibit high spatiotemporal variability and longer in winter when biospheric fluxes are small.

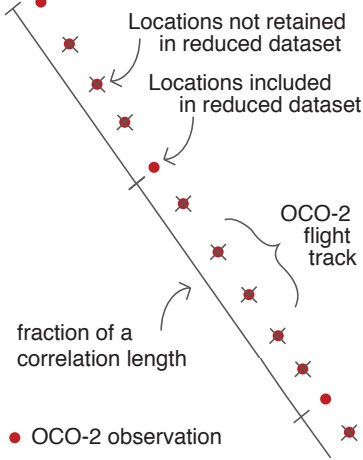

**Figure 2.** A schematic of the approach to data reduction proposed here. We estimate $XCO_2$ at one location per fraction of a correlation length along the satellite flight track, where the specific fraction must be chosen by the user. We subsequently estimate $XCO_2$ at each chosen location using ordinary kriging with a moving neighborhood.





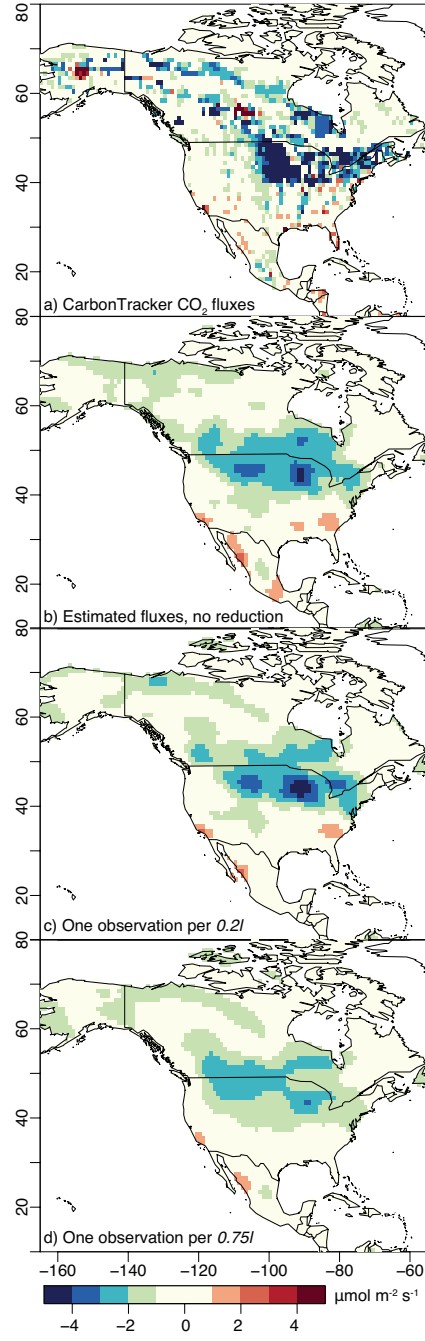

**Figure 3.** $CO_2$ fluxes estimated for the summer 2015 synthetic case study, averaged across the 6-week study window: (a) the synthetic $CO_2$ fluxes from NOAA's CarbonTracker estimate, (b) fluxes estimated from $XCO_2$ data with no reduction (6799 data points), (c) fluxes estimated from data reduced to one point per $0.2l$ (2263 data points), and (d) fluxes estimated from data reduced to one point per $0.75l$ (755 data points). The estimate with no reduction (b) and a reduction of $0.2l$ (c) reproduce broad, continental-scale spatial patterns in the synthetic fluxes (a), while the estimate with $0.75l$ reduction has lost spatial definition. Note that the inverse model here uses a non-informative prior, so any patterns in the flux estimates are informed by the observations, and not by any prior flux information.





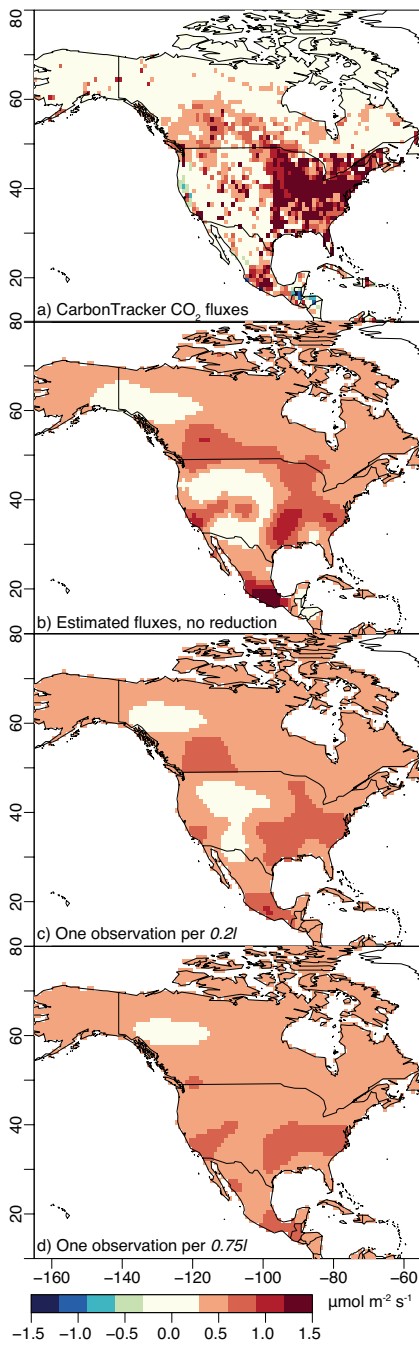

**Figure 4.** $CO_2$ fluxes estimated for the winter case study, analagous to Fig. 3. The fluxes estimated with modest data reduction (b, 1098 data points) reproduce the patterns in the flux estimate with no reduction (a, 4183 data points). By contrast, large data reduction (c, 251 data points) yields fluxes with little spatial variability.



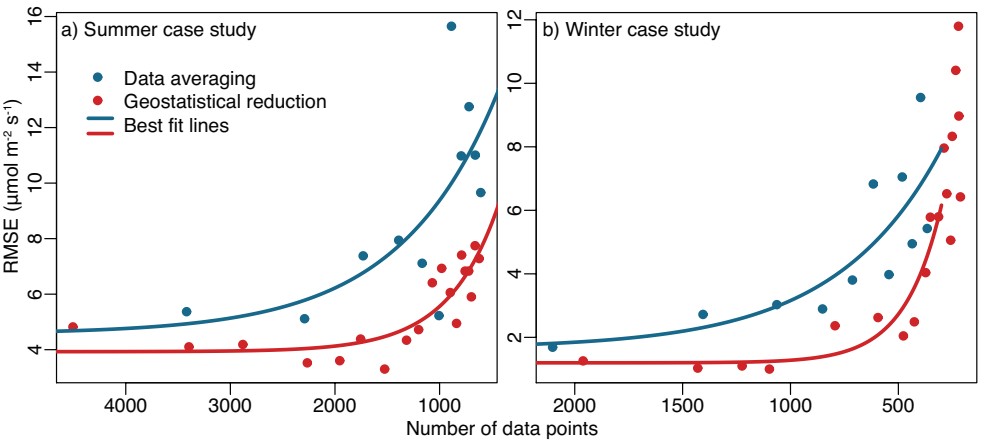

**Figure 5.** Root mean squared error (RMSE) of the fluxes estimated using data reduction relative to the fluxes estimated without data reduction. The figure displays results for the summer (a) and winter (b) case studies. Fluxes estimated using the data reduction approach proposed here have a lower RMSE relative to those estimated using the binning and averaging approach to data reduction.

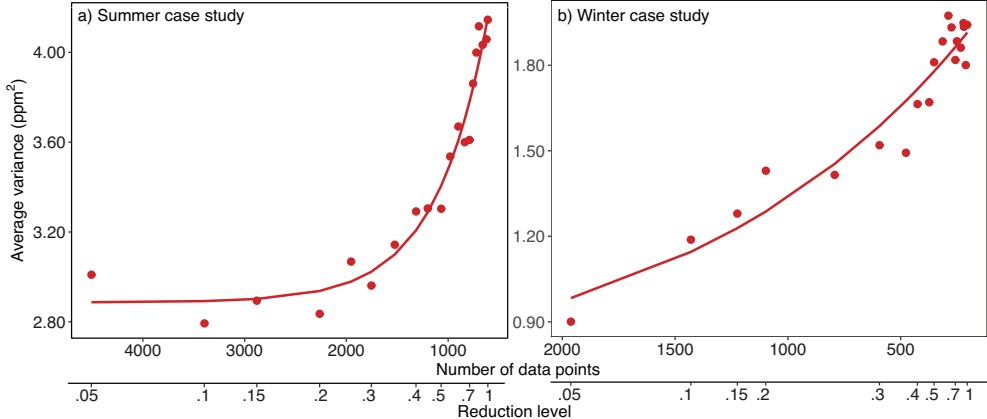

**Figure 6.** The amount of variance in the data that is lost through the process of data reduction for the summer (a) and winter (b) case studies. These plots provide a metric to help decide on an appropriate level of reduction and do not require costly runs of the inverse model to generate.