# Peer review of "Data reduction for inverse modeling: an adaptive approach v1.0"

_Geoscientific Model Development, 2020_

## Referee Comment (RC1) · Anonymous Referee #1 · 12 Jan 2021

Before I begin: What does v1.0 mean in the title of the paper. Is there an approach v2.0 ?? If there is one than I would like to see it

This manuscript requires revisions before publication. The manuscript adds another method to a series of existing methods for data reduction and authors are aware of this as they mention multiple approaches adopted in different disciplines to do this. Like clustering or any other method, a subjective criterion (through exploratory analysis is still needed to select observations) like a choice of an inflection point is required.

This is not a bad thing and the authors can choose magnitude of the slope to suggest an inflection point (Note the variance curve is monotonically increasing). I would like to also see if it makes a difference to use a correlation length obtained from summer months for one year and applied to observations of another year and in the second

stage select observations based on the variograms for the same year for which inversions are being performed. Thus, if it does not lead to any major difference in RMSE than these variograms can be precomputed for a selected year and applied for inversions for subsequent or previous years.

The section on variance criterion needs to be further elaborated. It is not clear, how the variances (is it variance in strict sense) are being computed. Is it the square of a difference between the kriged point and the support of the kriged point? Please demonstrate this with a small example within the context of Figure 2. Authors also need to elaborate whether the uncertainty from Ordinary Kriging was used in inversions or not. Furthermore, rather than using an exponential covariance structure for Ordinary Kriging why not use exponential + nugget model for doing kriging as this way you can have OCO-2 errors along the diagonal of the nugget and exponential structure can account for correlations in observations in space and/or time. This would allow Ordinary Kriging uncertainty to be directly included in data assimilation and/or inversions. I also wonder what would happen if you try to simultaneously use observations from multiple instruments and which due to different biases or viewing geometry leads to different values of XCO2 for a same location and completely contaminates the correlation length obtained from the variogram. This per se is not a problem with the methodology proposed here but can create problems in real-data applications of the methodology.

I would also like to see what kind of reduction happens if a variogram that accounts for both space and time is used to reduce the data. This can be checked within the context of a synthetic study and would become necessary in case of geostationary missions like GEOCARB that may have multiple overpasses for a same location over a single day.

Specific comments Line 40: replace particle with Lagrangian. Line 49: Why 2 seconds? How would you deal with it when you have simultaneous observations from multiple instruments? Line 54: Which vector? Line 59: What is the context of 30 days of wall-clock time (what kind of computer?) Line 62: Parallel computing architecture. Please

briefly elaborate especially within the context of 4D-Var and ensemble approaches
* * *

---

## Referee Comment (RC2) · Anonymous Referee #2 · 12 Jan 2021

Interactive comments on "Data reduction for inverse modeling: an adaptive approach v1.0"

The authors attempt to reduce computation cost of inverse modeling by reducing satellite data size based on different regional variability in satellite observations. They also attempt to use variability in reduced data to determine to which level the satellite data should be reduced. This work is interesting. And indeed, a need for data reduction in inverse modeling emerges with increasing number of satellite observations. However, the following issues need to be addressed before publication.

Major comments:

1. Do you consider the DOF of the satellite observations when you interpolate satellite

data? I wonder how much of the satellite DOF is lost after you interpolating the satellite observations. Would it make sense more if you determine the data reduction level with additional consideration of satellite DOF loss? In inverse modeling, averaging kernels are usually applied to remove impacts from satellite a priori. How do you deal with averaging kernels of OCO-2 XCO2 when you use kriging to interpolate satellite CO2 observations?

2. In your case studies, did you apply observation operator? If not, then you assume all satellite observations have identical DOF, which is not true for real satellite observations. Applying observation operator or not can make large difference on your top-down results. Thus, this can impact the evaluation of your data reduction algorithm.

3. Section 4.4: Since your goal is to reduce computation cost for inverse modeling, I wonder how large the difference is between the computation costs of your case studies (full data vs reduced data)? like the difference between computation costs of Figure 3 (b) and (c)? Also, did you try running your inverse modeling with full data, but using parallelization (e. g. 24 cores)? I wonder how large the computation cost would be compared to that of inverse modeling with reduced data but without parallelization.

Specific comments: 1. Line 55-62: Several problems here:

a. "The cost of running the forward and adjoint models usually increases as the number of observations increases, particularly for the model adjoint." The cost of running the forward and adjoint models would not increase as the number of observations increases. It is the cost of running observation operators that increases with increasing number of satellite observations.

b. "..a single run of GEOS-Chem adjoint model . . .30 days of . . .from the "lite" file." This sentence can not prove your point. I guess you want to prove that the cost of running one iteration of adjoint inversion with OCO-2 observations increase as the number of satellite observation increases. Then you need to compare the computation costs from at least two runs: one is assimilating fewer satellite observations, and the other is

assimilating more satellite observations.

c. "...many adjoint models for GHG applications are not designed to exploit parallel computing architecture." Can you pointing out what adjoint models do not use parallelization? As far as I know, most Eulerian chemical transport models and their adjoint models (if applicable) exploit parallelization, either OMP parallelization or MPI parallelization. For example, GEOS-Chem adjoint model exploits OMP parallelization.

d. "However, parallel versions are currently under development ... (Eastham et al., 2018)." What Eastham et al. (2018) developed is high-performance GEOS-Chem forward model (with MPI parallelization), not adjoint model.

2. Figure 4 caption: "...with modest data reduction (b, 1098 data points) ... with no reduction (a, 4183 data points). ... large data reduction (c, 251 data points) ...". I think it should be "...with modest data reduction (c, 1098 data points) ... with no reduction (b, 4183 data points). ...large data reduction (d, 251 data points) ..."
* * *

---

## Author Comment (AC1) · 30 Mar 2021

**Reply to reviewers**

We would like to thank the reviewers for their thoughtful comments on the manuscript. These ideas and suggestions have been very helpful to improve the quality and accuracy of the manuscript. Below, we have listed each of the reviewer's comments (in blue) and our replies.

**Reviewer #1:**

Before I begin: What does v1.0 mean in the title of the paper. Is there an approach v2.0 ?? If there is one than I would like to see it

This version numbering is a requirement of the journal and is not something that we, as authors, have control over (e.g., https://www.geoscientific-model-development.net/submission.html#manuscriptcomposition). When we first submitted the manuscript for publication, we did not include a version number in the title, but the editor requested that we add a version number as a stipulation of publication. There is no version 2, and we do not anticipate that there would be in the immediate future.

This is not a bad thing and the authors can choose magnitude of the slope to suggest an inflection point (Note the variance curve is monotonically increasing). I would like to also see if it makes a difference to use a correlation length obtained from summer months for one year and applied to observations of another year and in the second stage select observations based on the variograms for the same year for which inversions are being performed. Thus, if it does not lead to any major difference in RMSE than these variograms can be precomputed for a selected year and applied for inversions for subsequent or previous years.

   We would advise against computing the variogram for a selected year and applying it to subsequent years, and we make this recommendation for two reasons. First, Figs. 1 and S1 show the estimated correlation lengths for the synthetic and real data case studies, and the estimated correlation lengths exhibit substantial heterogeneity from one satellite flight track to another, even between flight tracks that are geographically proximal to one another. We hypothesize that variations in meteorology (in addition to variations in fluxes) can lead to substantial variability in the estimated correlation lengths (discussed in Sect. 4.1). Mesoscale and synoptic meteorology will change from day to day and certainly from year to year, so we feel it would be better to estimate the variograms for each flight track.
   Second, the computational cost of computing the variograms is not large, so the computational expense of estimating different variograms for different years is unlikely to be large. For example, we were able to compute the variograms for the summer case study (Fig. 1a) is less than one day on a personal laptop (a MacBook Air). We believe that it would be better to compute different correlation lengths for different years – both due to the possible effect of meteorology and due to the relatively small computational time required for the variogram analysis.
   We have added discussion of these points to Sect. 4.1 of the revised manuscript.

The section on variance criterion needs to be further elaborated. It is not clear, how the variances (is it variance in strict sense) are being computed. Is it the square of a difference between the kriged point and the support of the kriged point? Please demonstrate this with a small example within the context of Figure 2.

We have clarified this variance criteria in the revised manuscript. We first select all data points in the original $CO_2$ dataset that fall between two specific kriged points. We then calculate the variance of those selected points using the *var()* function in R. We repeat this procedure for each pair of kriged points in the model domain. We finally average these variances calculated across each pair of kriged points. This variance criteria thus shows the variance of the original dataset that is lost when we reduce the dataset to a smaller number of kriged data points. Some of this variance will undoubtedly be due to measurement error (e.g., the nugget), but some of this variance will be due to real variability in atmospheric $CO_2$.

Authors also need to elaborate whether the uncertainty from Ordinary Kriging was used in inversions or not. Furthermore, rather than using an exponential covariance structure for Ordinary Kriging why not use exponential + nugget model for doing kriging as this way you can have OCO-2 errors along the diagonal of the nugget and exponential structure can account for correlations in observations in space and/or time. This would allow Ordinary Kriging uncertainty to be directly included in data assimilation and/or inversions.

We have clarified this point in Sect. S3. We included a nugget in the variogram fitting process. Furthermore, we used the kriging uncertainties as the model-data mismatch errors in inverse modeling simulations using the reduced dataset.

The uncertainties in the ordinary kriging estimates are slightly lower than the errors in the individual synthetic $CO_2$ observations; this result is expected because ordinary kriging assimilates many observations to make the best possible $XCO_2$ estimate at each location (in this case at the observation locations). For example, in the summer case study, we added noise of 2 ppm (standard deviation) to the synthetic observations. On average, we estimated kriging uncertainties of 1.5 ppm. For the winter case study, we added noise of 1.5 ppm to the synthetic OCO-2 observations. On average, we estimated kriging uncertainties of 1.44 ppm. As a result, the model-data mismatch errors used in the inverse modeling simulations with the reduced dataset are slightly smaller than those used in the inverse modeling simulations with the original $CO_2$ dataset. Note that in the inverse modeling simulations using averaged data, we use the same model-data mismatch errors as in the inverse modeling simulations using the original $XCO_2$ dataset.

I also wonder what would happen if you try to simultaneously use observations from multiple instruments and which due to different biases or viewing geometry leads to different values of XCO2 for a same location and completely contaminates the correlation length obtained from the variogram. This per se is not a problem with the methodology proposed here but can create problems in real-data applications of the methodology.

We agree that this is an interesting question; the challenge of assimilating multiple different satellite datasets in an inverse model is very important yet very challenging. This topic is the focus of a multi-year NASA JPL MEaSUREs project (https://climatesciences.jpl.nasa.gov/co2measures/) led by Vineet Yadav. Several factors have made this task challenge to accomplish in practice. For example, GOSAT and OCO-2 exhibit contrasting biases in some regions of the globe, and one would presumably need to reconcile those biases before combining the different satellite datasets observations in an inverse modeling framework. Different satellite observations also have different spatial support, among other challenges. Differences in spatial support can create challenges when comparing or assimilating different datasets (e.g., Tadić and Michalak 2016).

We feel that this problem is both interesting and highly relevant but that tackling this problem is beyond the scope of the current manuscript. This topic could be an interesting challenge for a future paper, and the results could be informed by work currently being done as part of the NASA JPL

MEaSUREs project.

I would also like to see what kind of reduction happens if a variogram that accounts for both space and time is used to reduce the data. This can be checked within the context of a synthetic study and would become necessary in case of geostationary missions like GEOCARB that may have multiple overpasses for a same location over a single day.

In the present study, it made most sense to focus on spatial covariance for the OCO-2 case study. OCO-2 flight tracks have a 16-day repeat time, and the temporal covariance among different flight tracks is very small given the satellite's sampling pattern. Hence, our goal was to reduce the data spatially along each flight track.

Future geostationary satellite missions like GEOCARB will presumably sample at much higher spatial and temporal densities, and one would want to account for temporal covariances when reducing the data using an approach like that described in the current study.

It would be logistically and computationally difficult to construct new synthetic data simulations for GEOCARB in the context of the STILT model simulations used in this study. The GEOCARB satellite has not yet launched, so we do not yet have observations from GEOCARB. In addition, the number of STILT model simulations that would be required for this type of synthetic study would be computationally prohibitive. On one hand, the STILT model afforded enormous flexibility for the analysis presented in the current manuscript. We were able to run hundreds of inverse modeling simulations to test different setups for the data reduction algorithm and run inverse modeling simulations for numerous different levels of data reduction. If we wanted to run simulations for GEOCARB, we would need to switch models and use nested simulations with the GEOS-Chem model or TM5 model. Inverse modeling using the GEOS-Chem or TM5 adjoint requires iterating toward the solution and running the forward and adjoint of the GEOS-Chem or TM5 models at each iteration, a process that can take several days to several weeks (depending upon the specifics of the inverse model). These models therefore do not afford the same flexibility to test out large numbers of inverse modeling simulations, as was done in the present manuscript.

Line 40: replace particle with Lagrangian.

Done.

Line 49: Why 2 seconds?

We have clarified this point in the manuscript. This interval roughly corresponds with the spatial resolution of the meteorology used in STILT. The WRF-STILT simulations used in this study were generated as part of NOAA's CarbonTracker Lagrange program, a project funded by NASA's Carbon Monitoring System. The WRF simulations generated for STILT have a 10km resolution across the continental United States. The OCO-2 satellite travels approximately 10-km along its flight track every 2 seconds. As a result, STILT footprints were generated every 2 seconds along the flight track for CarbonTracker Lagrange.

How would you deal with it when you have simultaneous observations from multiple instruments?

This is a really good question but a tricky one. First and foremost, one would need a strategy to handle any offsets or biases between observations from different instruments (e.g., Tadić and Biraud 2020). Second, one would need to make an estimate of the different error variances and covariances of the observations from different instruments; the relative difference in accuracy/precision of the

instruments would be important for determining which to weight in a data reduction algorithm. Third, observations from different satellite instruments have different spatial support, and one would presumably want to account for those differences when combining multiple satellite datasets.

Line 54: Which vector?

We have clarified this reference in the manuscript.

Line 59: What is the context of 30 days of wall- clock time (what kind of computer?)

We have clarified this point in the revised manuscript.

Line 62: Parallel computing architecture. Please briefly elaborate especially within the context of 4D-Var and ensemble approaches.

We have re-written this paragraph in response to comments from reviewer #2. In the revised manuscript, we no longer refer to parallel computing architecture. Instead, we tried to reframe this paragraph in a way that is not specific to a particular model and is more generalizable to different inverse modeling approaches.

**Reviewer #2:**

Do you consider the DOF of the satellite observations when you interpolate satellite data? I wonder how much of the satellite DOF is lost after you interpolating the satellite observations. Would it make sense more if you determine the data reduction level with additional consideration of satellite DOF loss?

Some inverse modelers estimate the degree of freedom for signal (DOFS) in the inverse model. The DOFS are an estimate of the number of independent pieces of information provided by the observations for constraining the vector of unknown fluxes (i.e., the state vector) (e.g., Brasseur and Jacob 2017). This quantity is derived by summing the diagonal elements of the averaging kernel matrix calculated for the inverse problem. For many large problems, the calculation of the averaging kernel matrix can require substantial computational resources. For example, one way to calculate the averaging kernel matrix is to $(\mathbf{I} - \mathbf{V_s}\mathbf{Q}^{-1})$, where $\mathbf{Q}$ is the prior covariance matrix and $\mathbf{V_s}$ is the posterior covariance matrix. In the summer case study explored in this manuscript, $\mathbf{Q}$ has $1.05 \times 10^6$ rows and the same number of columns. $\mathbf{V_s}$ has the same dimensions. In addition, for variational or adjoint-based inversions, it is seldom possible to calculate $\mathbf{V_s}$ explicitly but rather must be approximated using Monte Carlo simulations or a reduced rank algorithm (e.g., Miller et al. 2020).

By contrast, our goal in this study is to create an algorithm for reducing greenhouse gas data and for evaluating the optimal level of data reduction without incurring burdensome computational costs. We considered DOFS as a criteria but ultimately did not opt to use this criteria due to the computational burdens involved in that calculation for large inverse problems.

In inverse modeling, averaging kernels are usually applied to remove impacts from satellite a priori. How do you deal with averaging kernels of OCO-2 XCO2 when you use kriging to interpolate satellite CO2 observations?

We did not try to remove the impact of the satellite prior on the satellite retrievals. This study is

primarily a synthetic data study, so it is not clear what the satellite prior would be in this context. We used kriging to estimate $XCO_2$ at locations that have OCO-2 observations. As such, we used pressure weighting functions and averaging kernels associated with those specific locations.

In your case studies, did you apply observation operator? If not, then you assume all satellite observations have identical DOF, which is not true for real satellite observations. Applying observation operator or not can make large difference on your top-down results. Thus, this can impact the evaluation of your data reduction algorithm.

We do apply an observation operator in the inverse model. The STILT model simulations used here are tailored to the individual locations, pressure weighting functions, etc. of the individual OCO-2 observations.

Section 4.4: Since your goal is to reduce computation cost for inverse modeling, I wonder how large the difference is between the computation costs of your case studies (full data vs reduced data)? like the difference between computation costs of Figure 3 (b) and (c)? Also, did you try running your inverse modeling with full data, but using parallelization (e. g. 24 cores)? I wonder how large the computation cost would be compared to that of inverse modeling with reduced data but without parallelization.

We have included additional text in the introduction and Sect. 4.4 to describe the computational costs associated with the STILT simulations in this study. The primary computational cost for the case studies described here is in the cost of generating WRF-STILT simulations that are used as inputs in the inverse model. The greater the level of data reduction, the fewer STILT simulations that would be required, and the smaller the computational cost of the overall forward and inverse modeling process. The important caveat, however, is that different types of atmospheric models entail different computational costs, and we have noted that in the revised version of the Introduction and Sect. 4.4 as well.

Different components of the forward and inverse modeling process can be done in parallel while other components of this process cannot be done in parallel. For example, one can distribute STILT simulations across many cores and/or nodes. Other components of the inverse model, however, cannot be done in parallel. For example, the statistical calculations often involve the inverse of several covariance matrices or the inverse of components of different covariance matrices. Those calculations cannot be done in parallel.

The cost of running the forward and adjoint models would not increase as the number of observations in- creases. It is the cost of running observation operators that increases with increasing number of satellite observations.

We have re-written this text and clarified this point in the revised manuscript. The computational cost of the forward and adjoint models themselves do not necessarily increase as the number of observations increase; we agree. The cost of the observation operator will increase with more observations. In addition, file I/O can be a bottleneck for some types of atmospheric models, and this cost increases as the number of observations increase. For example, to calculate the gradient of the inverse modeling cost function, the current $CO_2$ adjoint code for the GEOS-Chem model will read in the observations each time the adjoint model steps back in time. It does so as part of the process of applying the adjoint forcing. As a result, the current $CO_2$ adjoint code will read in the same set of observations many times during the course of a single model run.

"..a single run of GEOS-Chem adjoint model . . .30 days of . . .from the "lite" file." This sentence can

not prove your point. I guess you want to prove that the cost of running one iteration of adjoint inversion with OCO-2 observations increase as the number of satellite observation increases. Then you need to compare the computation costs from at least two runs: one is assimilating fewer satellite observations, and the other is assimilating more satellite observations.

We have updated this paragraph and compare the computational cost of using the full lite file versus 10-second data averages that are currently being used in the OCO-2 model inter-comparison project. Furthermore, we have revised this statement to compare the total computing time to calculate the inverse modeling cost function and gradient in these two cases.

". . .many adjoint models for GHG applications are not designed to exploit parallel computing architecture." Can you pointing out what adjoint models do not use parallelization? As far as I know, most Eulerian chemical transport models and their adjoint models (if applicable) exploit parallelization, either OMP parallelization or MPI parallelization. For example, GEOS-Chem adjoint model exploits OMP parallelization.

We have re-written this paragraph. For the $CO_2$ adjoint code in GEOS-Chem, the largest computational bottleneck appears to come in file I/O – specifically reading in the observations. The current adjoint code will read in the observations and apply the adjoint forcing at each time step of the adjoint model. For the $CO_2$ adjoint, the default time step is one hour, so the adjoint model will step one hour backward in time, the code will read in the observations, and the process repeats. This file I/O is not done in parallel in the current adjoint code.

In the revised manuscript, we have removed specific references to parallelization and have revised the text to be more general and not as model-specific. Instead, we state that the observation operator and file I/O can be time-consuming for inverse modeling using variational or adjoint-based inverse models. We also point out that several recent satellite-based inverse modeling efforts of $CO_2$ use satellite data that have been reduced in some way. For example, the OCO-2 model inter-comparison project employs 10-second averages of OCO-2 observations (e.g., Crowell et al. 2019).

"However, parallel versions are currently under development . . . (Eastham et al., 2018)." What Eastham et al. (2018) developed is high-performance GEOS-Chem forward model (with MPI parallelization), not adjoint model.

We have removed this sentence from the revised manuscript as part of the revisions described under the previous point.

Figure 4 caption: ". . .with modest data reduction (b, 1098 data points) . . . with no reduction (a, 4183 data points). . . . large data reduction (c, 251 data points) . . .". I think it should be ". . .with modest data reduction (c, 1098 data points) . . . with no reduction (b, 4183 data points). . . .large data reduction (d, 251 data points) ..."

Thank you for catching this error. We have fixed this error in the revised version of the manuscript.

**References:**

Brasseur, G.P., Jacob, D.J. (2017). Modeling of Atmospheric Chemistry. United Kingdom:Cambridge University Press. Doi:10.1017/9781316544754.

Crowell, S., Baker, D., Schuh, A., Basu, S., Jacobson, A. R., Chevallier, F., Liu, J., Deng, F., Feng, L.,

McKain, K., Chatterjee, A., Miller, J. B., Stephens, B. B., Eldering, A., Crisp, D., Schimel, D., Nassar, R., O'Dell, C. W., Oda, T., Sweeney, C., Palmer, P. I., and Jones, D. B. A.: The 2015–2016 carbon cycle as seen from OCO-2 and the global in situ network, Atmos. Chem. Phys., 19, 9797–9831, https://doi.org/10.5194/acp-19-9797-2019, 2019.

Liu, J., Baskaran, L., Bowman, K., Schimel, D., Bloom, A. A., Parazoo, N. C., Oda, T., Carroll, D., Menemenlis, D., Joiner, J., Commane, R., Daube, B., Gatti, L. V., McKain, K., Miller, J., Stephens, B. B., Sweeney, C., and Wofsy, S.: Carbon Monitoring System Flux Net Biosphere Exchange 2020 (CMS-Flux NBE 2020), Earth Syst. Sci. Data, 13, 299–330, https://doi.org/10.5194/essd-13-299-2021, 2021.

Tadić, Jovan M., and Anna M. Michalak.: On the effect of spatial variability and support on validation of remote sensing observations of $CO_2$. Atmospheric Environment, 132, 309-316, doi:10.1016/j.atmosenv.2016.03.014, 2016.

Tadić, Jovan M., and Sébastien C. Biraud: Identification of Bias in Satellite Measurements Using Its Geospatial Properties. IEEE Geoscience and Remote Sensing Letters, doi:10.1109/LGRS.2020.3015174, 2020.

---

## Author Response (AR2)

**Reviewer replies**

We thank the reviewers for their thoughtful suggestions on the manuscript. Below, we have listed each of the reviewer's comments and have listed replies in blue below each comment.

It's still not clear to me whether or not the authors have correctly applied the observation operator in their inverse modeling. Here in this study, the observation operator should include application of satellite a priori, averaging kernels, pressure weighting function from OCO-2 level2 data files. Please refer to Cogan et al. (2012) for the application of observation operator in comparison between model and satellite XCO2 observations. The application of averaging kernel is very important when comparing model and satellite observations, as this can reduce the impact from satellite a priori. In your synthetic cases, you can also generate synthetic satellite data using model simulation and satellite a priori, averaging kernels, pressure weighting function from OCO-2 level2 data files. Correctly using observation operator or not can impact your top-down CO2 fluxes from both full-data inversion and reduced-data inversion. Thus this can impact the evaluation of your data reduction algorithm.
I recommend publication after the authors clarifying this part and addressing the following questions/comments.

We followed a similar approach to the observation operator as in Cogan et al. (2012). According to Cogan et al. (2012):
$$\mathbf{X_{CO2}} = \mathbf{h}^T\mathbf{x_a} + \mathbf{h}^T\mathbf{A}(\mathbf{x} - \mathbf{x_a}) \qquad (1)$$
where $\mathbf{X_{CO2}}$ is the retrieved $CO_2$ observation, $\mathbf{h}$ is the pressure-weighting function, $\mathbf{x_a}$ is the *a priori* estimate of the $CO_2$ profile, $\mathbf{A}$ is the averaging kernel, and $\mathbf{x}$ is the true $CO_2$ profile. Equation 1 can also be re-arranged into two components – information contributed by the prior and by the satellite observation (e.g., Brasseur and Jacob, ch. 11):
$$\mathbf{X_{CO2}} \quad = \mathbf{X_{prior}} + \mathbf{X_{satellite}} \qquad (2)$$
$$\mathbf{X_{prior}} \quad = \mathbf{h}^T\mathbf{x_a}(1 - \mathbf{A}) \qquad (3)$$
$$\mathbf{X_{satellite}} \quad = \mathbf{h}^T\mathbf{A}\mathbf{x} \qquad (4)$$
A common approach in existing satellite-based inverse modeling studies is to either (a) apply Eq. 1 (or Eq. 2) to the atmospheric model outputs before comparing against the retrieved $CO_2$ observations, or (b) subtract the component that is likely due to the prior from the retrieved $CO_2$ observations ($\mathbf{X_{CO2}}$ - $\mathbf{X_{prior}}$ = $\mathbf{X_{satellite}}$) and apply Eq. 4 to the model outputs (e.g., Frankenberg et al. 2006, Bergamaschi et al. 2007, Basu et al. 2013, Saeki et al. 2013). We use the latter approach in the real data simulations:
$$\mathbf{z} = \mathbf{X_{CO2}} - \mathbf{X_{prior}} - \mathbf{h}^T\mathbf{A}\mathbf{b} \qquad (5)$$
where $\mathbf{z}$ are the processed observations used in the inverse model and $\mathbf{b}$ is the $CO_2$ background or clean air boundary condition.

An exception is that we did not include $\mathbf{x_a}$ in the synthetic data simulations – because it cancels out. Let us suppose that we include an *a priori* $CO_2$ profile in the synthetic data. We would generate the synthetic satellite data ($\mathbf{X_{CO2}}$) using the equation outlined above: $\mathbf{X_{CO2}} = \mathbf{X_{prior}} +$ $\mathbf{X_{satellite}} + \mathbf{h}^T\mathbf{A}\mathbf{b}$. Before running the inverse model, we would apply Eq. 5 and subtract $\mathbf{X_{prior}}$ from the synthetic satellite. As a result, the term $\mathbf{X_{prior}}$ cancels out.

Specific comments:

Line 52-53: "Rather, these models are often used to calculate the product of H or HT and a vector (e.g., a vector of estimated CO2 fluxes)". Here, a vector of estimated CO2 fluxes is not an appropriate example. Usually, the vector is the gradients of your objective function with respect to simulated targeted species, here in your study, simulated CO2 profiles.

We have removed this example from the manuscript. We originally added an example because it was requested by a different reviewer. Chemical transport models are often used to calculate the product of $\mathbf{H}$ and a vector of estimated $CO_2$ fluxes. By contrast, the adjoint of a chemical transport model is often used to calculate the product of $\mathbf{H}^T$ and a vector, and that vector can vary depending upon the application and specific inverse modeling approach used. For example, that vector will differ depending on whether one uses a gradient-based method for minimizing the inverse modeling cost function (as mentioned by the reviewer above) or whether using a different approach like the minimum residual method (e.g., Saibaba and Kitanidis 2012). To avoid any confusion, we have removed the text in parentheses.

Line 55-56: "The model output must be interpolated to the locations of the observations, often referred to as the observation operator." Partially correct. The key part of the observation operator, here in your study, is the application of satellite a priori, averaging kernels and pressure weighting function.

We have removed the term "observation operator" from the text to avoid confusion.

Line 59-66: I wonder how different are the gradients at the first iteration between using reduced data and using full data, in terms of spatial distribution and magnitude?

We did not calculate gradients as part of the inverse modeling simulations in this manuscript. We found the minimum of the inverse modeling cost function directly (aka, analytically) instead of using an iterative minimum-finding algorithm. We have added text to Sect. S3 that provides additional detail on the equations used to minimize the inverse modeling cost function.

**References:**

Basu, S., Guerlet, S., Butz, A., Houweling, S., Hasekamp, O., Aben, I., Krummel, P., Steele, P., Langenfelds, R., Torn, M., Biraud, S., Stephens, B., Andrews, A., and Worthy, D. (2013). Global $CO_2$ fluxes estimated from GOSAT retrievals of total column $CO_2$, Atmos. Chem. Phys., 13, 8695–8717, doi:10.5194/acp-13-8695-2013.

Bergamaschi, P., et al. (2007), Satellite chartography of atmospheric methane from SCIAMACHY on board ENVISAT: 2. Evaluation based on inverse model simulations, J. Geophys. Res., 112, D02304, doi:10.1029/2006JD007268.

Brasseur, G.P., Jacob, D.J. (2017). Modeling of Atmospheric Chemistry. United Kingdom: Cambridge University Press, doi:10.1017/9781316544754.

Frankenberg, C., Meirink, J. F., Bergamaschi, P., Goede, A. P. H., Heimann, M., Körner, S., Platt, U., van Weele, M., and Wagner, T. (2006), Satellite chartography of atmospheric methane

from SCIAMACHY on board ENVISAT: Analysis of the years 2003 and 2004, J. Geophys. Res., 111, D07303, doi:10.1029/2005JD006235.

Saeki, T., Maksyutov, S., Saito, M., Valsala, V., Oda, T., RJ, A., ... & Yokota, T. (2013). Inverse modeling of $CO_2$ fluxes using GOSAT data and multi-year ground-based observations. Sola, 9, 45-50, doi:10.2151/sola.2013-011.

Saibaba, A. K., and Kitanidis, P. K. (2012), Efficient methods for large-scale linear inversion using a geostatistical approach, Water Resour. Res., 48, W05522, doi:10.1029/2011WR011778.